# Research on the Residual Vibration Suppression of Delta Robots Based on the Dual-Modal Input Shaping Method

**Zhongfeng Guo, Jianqiang Zhang \* and Peisen Zhang**

Liaoning Provincial Key Laboratory of Intelligent Manufacturing and Industrial Robots, Shenyang University of Technology, Shenyang 110870, China
\* Correspondence: zhang_jq@smail.sut.edu.cn

**Abstract:** The Delta robot is a high-speed and high-precision parallel robot. When it is in function, the end effector generates residual vibration, which reduces the repeat positioning accuracy and positioning efficiency. The input shaping method has previously been shown to suppress the residual vibration of the robot, but the vibration suppression effect of the single-modal input shaper is not good for the delta robot, which has multiple dominant modes for the residual vibration. To solve this problem, this paper proposes an effective method for residual vibration suppression of Delta robots based on dual-modal input shaping technology. Firstly, the modal analysis of the Delta robot is performed using finite element software, and the dominant modal of its residual vibration is determined. Secondly, six dual-modal input shapers are designed according to the obtained modal parameters. Finally, Simulink is used for simulation analysis to verify the robustness and vibration suppression performance of the designed six dual-modal input shapers and traditional single-modal input shapers. The simulation results show that the designed ZVD-EI dual-modal input shaper has good robustness, can effectively suppress the residual vibration of the Delta robot, and can effectively improve the repetitive positioning accuracy and work efficiency of the Delta robot when it is running at high speed.

**Keywords:** Delta robot; vibration suppression; modal analysis; multimodal; input shaping

## 1. Introduction

Parallel robots are widely used in electronics, food, pharmaceuticals, and other industrial applications due to their high speed, high accuracy, and excellent motion performance. Delta robots are a typical example, first introduced by Clavel [1] in the 1980s. Since then, researchers have focused on DELTA parallel robots, and many valuable studies have been proposed [2–5]. However, key components must be lightweight to achieve high-speed motion, which can lead to residual vibrations at high speeds and reduce the repeatability and positioning efficiency of the robot's end effector. Therefore, residual vibration suppression in Delta robots has become one of the hot topics of research.

At present, research on vibration suppression for Delta robots mainly includes the use of trajectory planning methods, adaptive robust control methods, and input shaping methods. In the literature [6], to reduce mechanism vibration and speed fluctuations and ensure smooth transitions in operation, the feasibility of the real-time path smoothing method is verified through experiments. In the literature [7], for the requirements of dynamic pick-and-place high-speed stability of Delta robots, multi-segment polynomial designs such as 4-3-4, 3-5-3, and 5-3-5 are used to establish a non-linear motion trajectory planning model, and the optimal solution of trajectory planning is obtained to verify the effectiveness in controlling mechanism vibration. In the literature [8], a trajectory planning method is proposed to consider both the motion smoothness and dynamic stress of the Delta robot. A modified fifth-order b-sample method is used for sensitivity analysis and normalized time factor optimization, and the results show that this method can improve

the motion smoothness while reducing the dynamic stress. In the literature [9], an adaptive robust control method based on a fuzzy dynamics model is established for the Delta robot with unknown dynamics parameters, residual vibration disturbances, and other factors, using the uncertainty of the fuzzy description; the uncertainty information is estimated through an adaptive mechanism; and finally, the effectiveness of the control method is verified through simulation.

However, the use of trajectory planning methods or adaptive robust control can only suppress the process vibration of Delta robots to a certain extent when the robot is moving at high speeds and with high acceleration. To improve the repetitive positioning accuracy and positioning efficiency of the robot's end effector, the residual robot vibration must be effectively suppressed. The input shaping method [10,11] is a commonly used feed-forward-based vibration suppression method, where the actual input model of the controlled system is obtained by convolving the input signal with a series of pulse sequences to eliminate the residual vibrations generated by the system. The method was originally applied to overhead cranes [12], where the application of input shaping algorithms to the control system was effective in reducing residual oscillations in the crane boom and improving efficiency. The technique has now been used in various industrial robots [13,14].

The most fundamental component of input shapers is the ZV (Zero-Vibration) input shaper. In order to design an effective ZV input shaper, it is usually necessary to derive the exact intrinsic frequency and damping ratio of the system; however, due to the uncertainties in modeling and simulation, it is not possible to obtain completely accurate modal parameters, so it is necessary to ensure the robustness of the designed input shaper. Researchers have proposed different input shapers, such as the ZVD (Zero-Vibration and Derivative) input shaper and EI (Extra-Insensitive) input shaper, based on the ZV input shaper [15]. In the case of multimodal-dominated systems, multiple unimodal input shapers are used for convolution to obtain multi-modal input shapers for residual vibration suppression of multi-modal systems [16–18]. Initially, the input-shaping method was mainly used in linear constant systems, but its application has gradually been extended to non-linear systems and has been more extensively studied in residual vibration suppression of parallel robots [19–21].

The Delta robot is a typical multi-modal dominated non-linear system with residual-vibration-dominated modal parameters that vary greatly from position to position in the workspace. This makes the traditional single-modal input shaper described above less effective in suppressing the residual vibration of the Delta robot. Therefore, an effective input shaper needs to be designed to provide good robustness and vibration suppression performance of the Delta robot throughout the working space. Inspired by the existing results, this paper proposes a dual-modal input shaper-based residual vibration suppression method for Delta robots based on the multi-dominant modal characteristics of Delta robotic systems to improve the robustness and residual vibration suppression performance of conventional single-modal input shapers in Delta robotic applications.

The main contributions of this paper can be summarized as follows:(a) Using ANSYS modal analysis, the two-order modes that play a dominant role in the residual vibration of the Delta robot were identified and the modal parameters were obtained for the subsequent design of the input shaper. (b) The six dual-modal input shapers were designed for the system characteristics of the Delta robot, and the input shapers were built in Simulink. (c) The robustness of the six bimodal input shapers was evaluated by simulation because the dominant modal parameters of the Delta robot vary considerably at different locations in the workspace. After giving certain error values to the modal parameters, the robustness and residual vibration suppression performance of the six dual-modal input shapers were analyzed and it was found that the established ZVD-EI dual-modal input shaper could guarantee both a short adjustment time and good vibration suppression performance.

## 2. Model of the Delta Robot

The 3D model of the 4-DOF Delta robot is shown in Figure 1. This robot is mainly composed of a fixed platform, a moving platform, an intermediate shaft, three driving arms, and three driven arms. Each driven arm contains a set of parallelogram branch chains. Driven by three driving arms, the moving platform can realize high-speed movement of the X–Y–Z axis. Driven by the motor connected to the intermediate shaft, the end effector under the moving platform can realize Z-axis rotation. The specific structure of each component is shown in Figure 2.

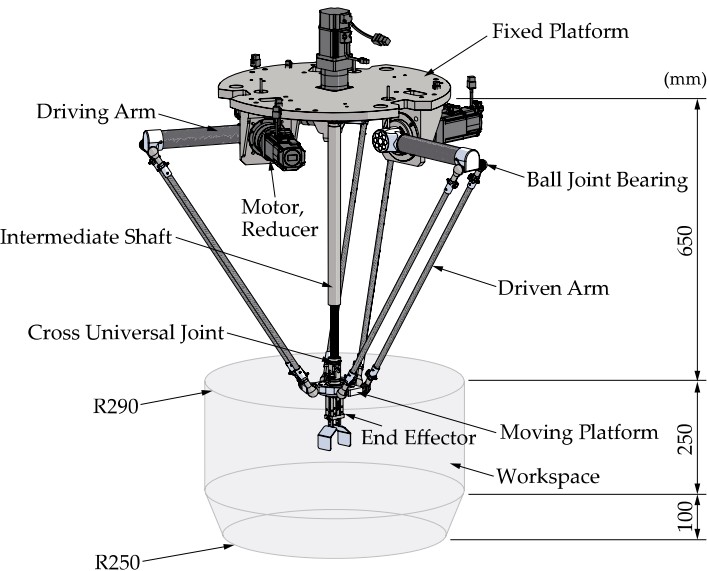

**Figure 1.** Delta robot 3D model.

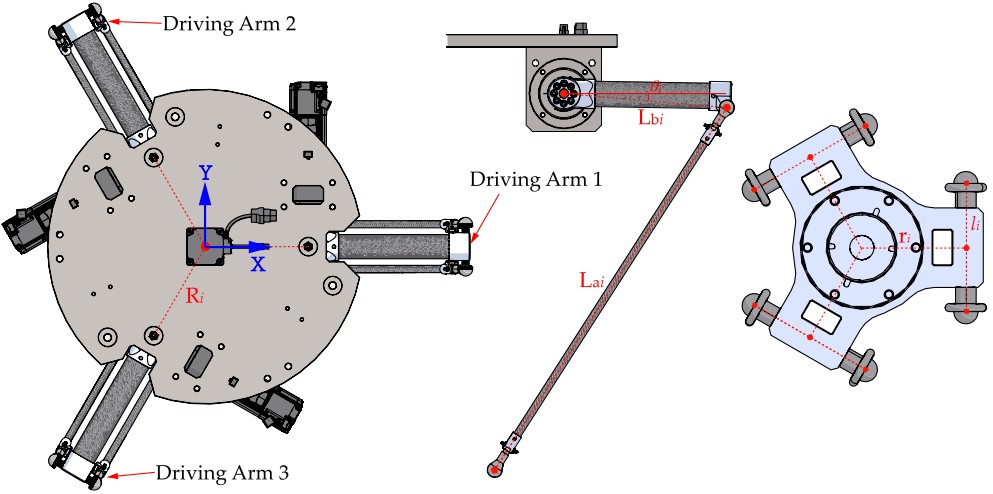

**Figure 2.** Notation used in the model of the Delta robot.

## 3. Elastic Dynamics Modeling and Modal Analysis

The Delta robot is a time-varying system with multiple modals, and its system modal parameters change according to the position of the robot. In order to design a suitable input shaper, it is first necessary to establish an elastodynamic model that accurately estimates the inherent characteristics of the Delta robot and then determine the modal that plays a dominant role in the residual vibration of the end effector [21].

### 3.1. Elastic Dynamic Modeling

Considering that the structure of the Delta robot is complex and there are many joints and hinges, to accurately estimate the modal parameters of the Delta robot, a 3D model of the Delta robot is established in Solidworks, and the overall finite element model is established by using ANSYS software after proper simplification. The geometrical and dynamic parameters of the Delta robot used in this paper are reported in Tables 1 and 2. Known system parameters: both the main body of the driving arm and the driven arm are made of carbon fiber ($\rho = 1800 \text{ kg/m}^{-3}$), and the connection part is made of aluminum alloy ($\rho = 2770 \text{ kg/m}^{-3}$). The diameter of the driving arm is 40 mm, the wall thickness is 5 mm, the diameter of the driven arm is 12 mm, and the wall thickness is 3 mm. The moving platform and the intermediate shaft are made of aluminum alloy.

**Table 1.** Geometrical parameters of the Delta robot.

| Description | Notation | Value |
|---|---|---|
| Radius of the fixed platform | $R_i$ | 0.17 m |
| Radius of the moving platform | $r_i$ | 0.06 m |
| Length of the $i$-th driven arm | $L_{ai}$ | 0.68 m |
| Length of the $i$-th driving arm | $L_{bi}$ | 0.26 m |
| Width of the branch chain of the $i$-th driven arm | $l_i$ | 0.074 m |
| Angle of the $i$-th active joint | $\theta_i$ | $-35°{\sim}75°$ |

**Table 2.** Dynamic parameters of the Delta robot.

| Description | Value |
|---|---|
| Shear modulus of elasticity of the driving arm and driven arm | $9.0 \times 10^9 \text{ N/m}^2$ |
| Young's modulus of the driving arm and driven arm | $2.3 \times 10^{10} \text{ N/m}^2$ |
| Poisson coefficient | 0.2 |
| Friction coefficient | 0.11 |
| Mass of the $i$-th driving arm | 0.693 kg |
| Mass of the $i$-th driven arm | 0.175 kg |
| Mass of moving platform | 0.52 kg |
| Load mass | 1 kg |

Due to the different positions and attitudes of the Delta robot when it is in function, its mass matrix and stiffness matrix are also different. To facilitate the modal analysis, a specific trajectory is selected to analyze the modal changes of the robot. Given the following trajectory of the end effector, the position points $P_0$, $P_1$, $P_2$, $P_3$, $P_4$, and $P_5$ are selected as shown in Figure 3. The position coordinates of its position point in the global coordinate system are given in Table 3. The origin o of the global coordinate system is the center of the plane formed by the output axes of the three sets of reducers of the robot, where $P_1P_2$ and $P_4P_5$ are circular arc segments, and the rest of the trajectories are straight-line segments. The motion trajectory in Figure 3 is calculated by the quintic polynomial trajectory planning method, the total length of the motion trajectory is 0.6048 m, and the motion time is 0.25 s. The position, velocity and acceleration curves of Delta robot in joint space and Cartesian space can be obtained, as shown in Figure 4.

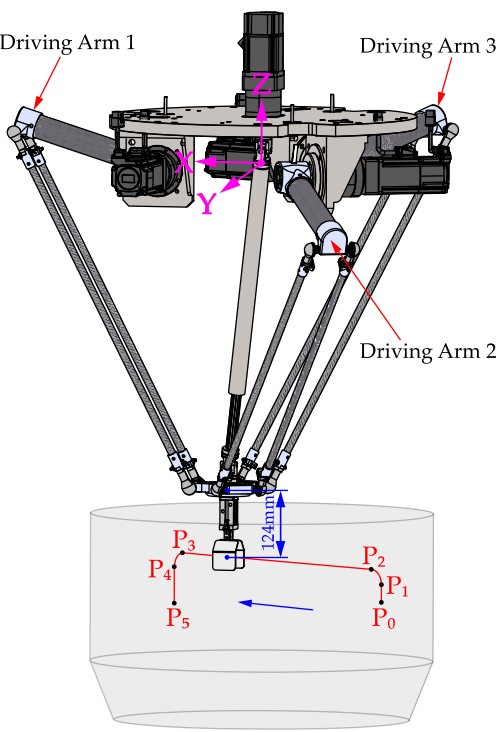

**Figure 3.** Trajectory of the Delta robot.

**Table 3.** Position coordinates.

| Position Point | X (mm) | Y (mm) | Z (mm) |
|:---:|:---:|:---:|:---:|
| $P_0$ | −200 | 0 | −740 |
| $P_1$ | −200 | 0 | −710 |
| $P_2$ | −182 | −9 | −670 |
| $P_3$ | 182 | −191 | −670 |
| $P_4$ | 200 | −200 | −710 |
| $P_5$ | 200 | −200 | −780 |

The simulation steps are as follows:

1. Establish a parametric geometric model of the Delta robot according to actual needs, as shown in Figure 1.
2. Take the Delta robot model shown in Figure 1 as the verification object, simplify its geometric model, and import it into ANSYS workbench in the x_t standard format.
3. Import the component features (material and unit type) and connection features (joint contact type/fixed support) into the finite element software, perform mesh division, and establish the finite element model of the robot.

### 3.2. Delta Robot Modal Analysis

When analyzing the elastic dynamics of the Delta robot system, the fixed platform and the kinematic pair are regarded as rigid elements, and the deformation during the movement is not considered; the driving arm, driven arm, and moving platform are regarded as flexible elements. Therefore, the Delta robot is meshed in the ANSYS workbench, and the first six order modals of the center position coordinates of the Delta robot end effector at the above six position points are calculated. The natural frequencies of each order of the system are shown in Figure 5. It can be seen that the natural frequency of the robot gradually increases with the increase in the modal order in any position. Table 4 describes the modal shape of the robot at each position point. In the first two order modals, the modal shape of the robot is the torsion of the moving platform along the XY plane. Obviously, the torsion along the XY plane has the greatest impact on the robot and is most likely to cause the end

effector vibration. In the design of the input shaper below, the influence of the first two order modals on the residual vibration of the Delta robot should be considered at the same time. Therefore, the first two order modes are taken as the dominant modal of the residual vibration of the Delta robot under this specific trajectory. Figures 6 and 7 show the first two order modals' shape contours of each location point.

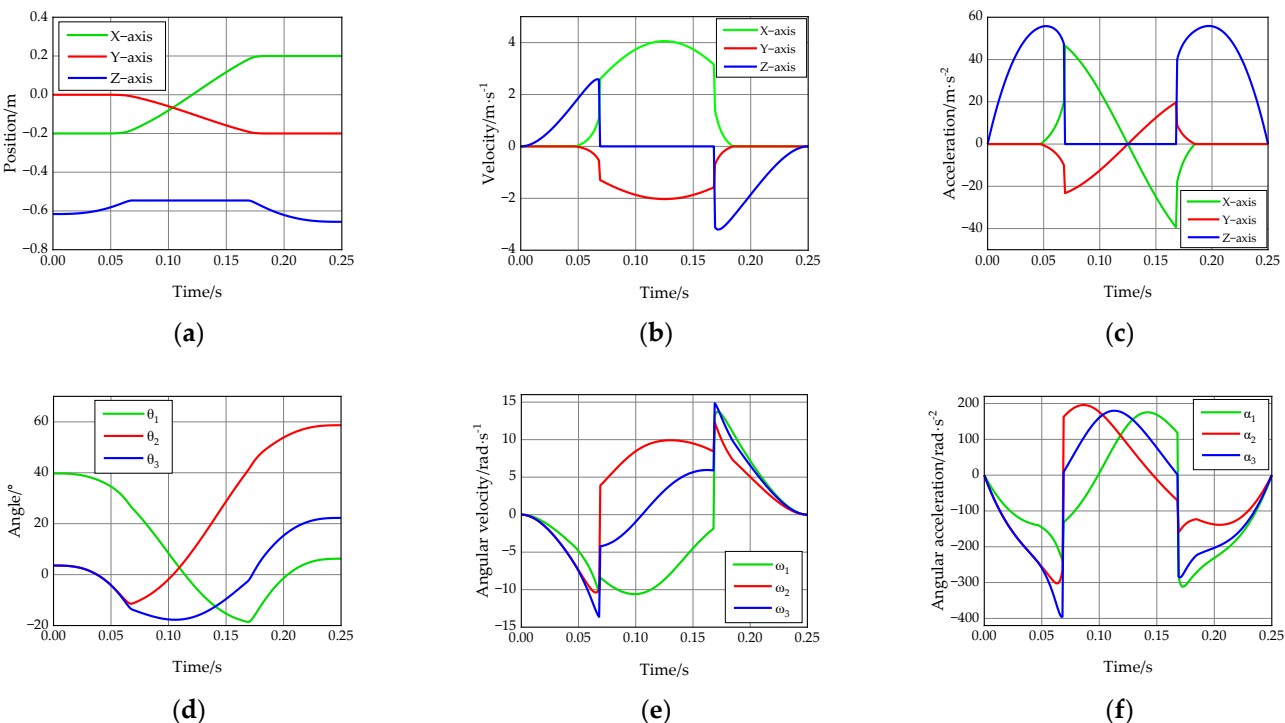

**Figure 4.** Physical quantity of the Delta robot under the motion trajectory: (**a**) position curve of the robot in the Cartesian space; (**b**) velocity curve of the robot in the Cartesian space; (**c**) acceleration curve of the robot in the Cartesian space; (**d**) angle curve of the robot in the joint space; (**e**) angular velocity curve of the robot in the joint space; (**f**) angular acceleration curve of the robot in the joint space.

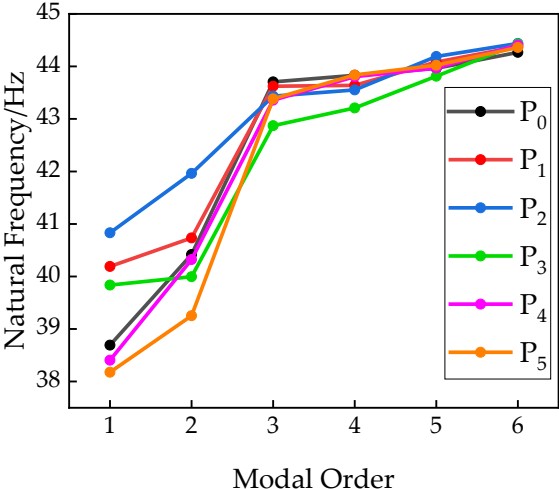

**Figure 5.** First six modals of each position point.

**Table 4.** Description of point vibration shapes of the robot at each position.

| Order | Maximum Frequency/Hz | Minimum Frequency/Hz | Modal Shape Description |
|---|---|---|---|
| 1 | 40.832 | 38.177 | The moving platform is twisted along the XY plane |
| 2 | 41.963 | 39.253 | The moving platform is twisted along the XY plane |
| 3 | 43.706 | 42.871 | Deflection of a driven arm chain along its parallelogram plane |
| 4 | 43.836 | 43.208 | The deflection deformation of two driven arm chains along their parallelogram planes |
| 5 | 44.19 | 43.813 | The deflection deformation of two driven arm chains perpendicular to their parallelogram plane |
| 6 | 44.437 | 44.268 | Two driven arm chains deflect along its parallelogram plane, and one driven arm chain deflects perpendicularly to its parallelogram plane |

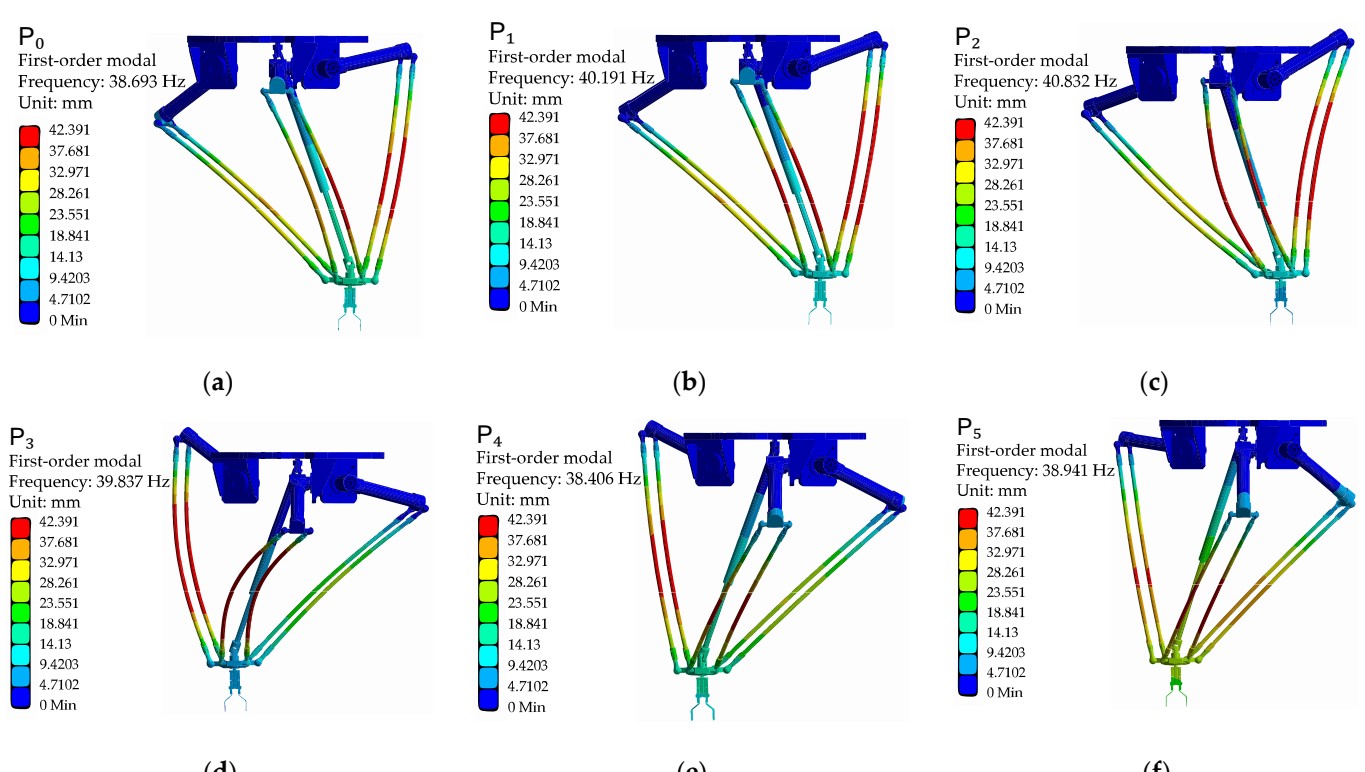

**Figure 6.** First-order modal of each position point: (**a**) mode shape contour at point $P_0$; (**b**) mode shape contour at point $P_1$; (**c**) mode shape contour at point $P_2$; (**d**) mode shape contour at point $P_3$; (**e**) mode shape contour at point $P_4$; (**f**) mode shape contour at point $P_5$.

According to the cloud diagram analysis of the first six order modals at each point of the Delta robot, the deformation is mainly located on the driven arm. This is because the driven arm is mainly made of carbon fiber material. The driven arm is a slender rod, and its aspect ratio is the same as that of the driving arm; it is more flexible and is more likely to cause vibration deformation. Therefore, when optimizing the vibration reduction in the Delta robot, we should focus on the dynamic characteristics of the driven arm.

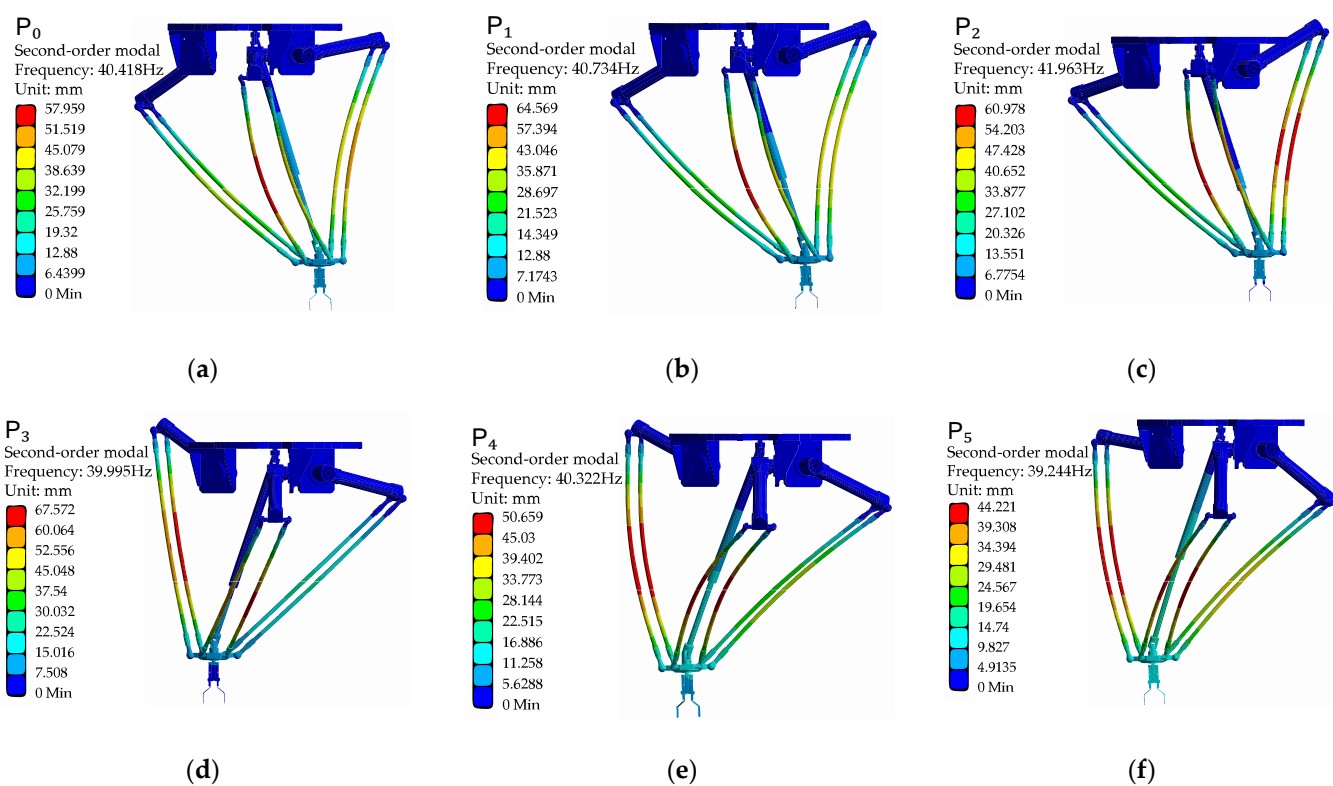

**Figure 7.** The second-order modal of each position point: (**a**) mode shape contour at point $P_0$; (**b**) mode shape contour at point $P_1$; (**c**) mode shape contour at point $P_2$; (**d**) mode shape contour at point $P_3$; (**e**) mode shape contour at point $P_4$; (**f**) mode shape contour at point $P_5$.

## 4. Design of the Input Shaping Controller

### 4.1. Principle of Input Shaping Technology

Input shaping is a vibration suppression algorithm. It convolves a group of pulse signals with the system input signal to obtain the actual input signal of the control system to avoid unnecessary vibration in the system. The input shaping method is based on the posicast principle [22,23], that is, sending out the first pulse produces a dynamic response in the system. After a certain period of time, a second pulse is introduced. If the second pulse has the correct time and amplitude, it can cancel the response produced by the first pulse. The principle is shown in Figures 8 and 9. Input shaping is a typical feed-forward control method, which can be effectively applied to the oscillation suppression of the system. The frame diagram of the input shaper system is shown in Figure 10. Compared with the closed-loop vibration suppression method, input shaping does not require real-time measurement of the vibration deformation of the system [24]. The input shaper only needs to be designed to identify the natural frequency and damping ratio of the system and does not require accurate modeling of the system; it is also suitable for complex and difficult-to-model structures.

However, reference signals for robots typically consist of a reference position, possibly in conjunction with other reference signals such as the reference velocity, reference torque, etc. Thus, it is necessary to modify these reference signals such that they will not introduce vibrations in the system [17].

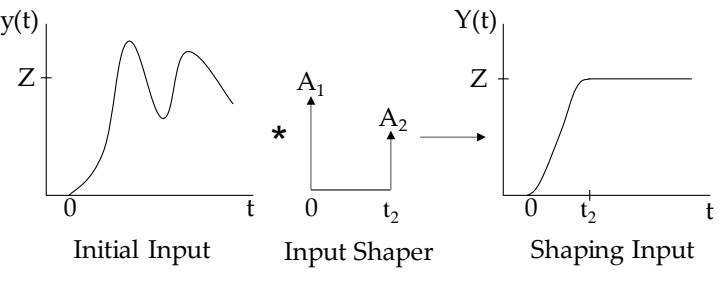

**Figure 8.** Single-modal input shaping control principle.

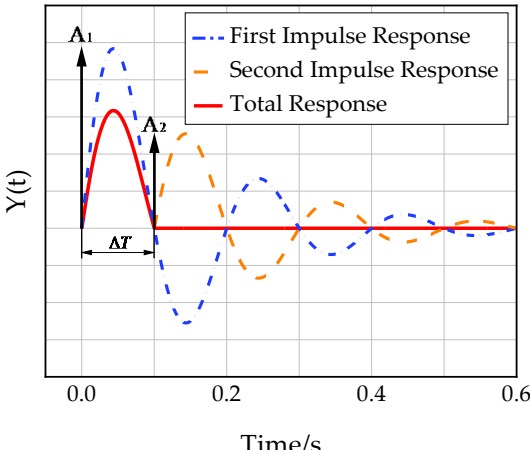

**Figure 9.** ZV input shaper impulse response.

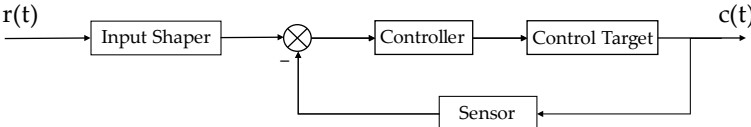

**Figure 10.** Frame diagram of the input shaping system.

The Delta robot can be modeled as an underdamped second-order system with a transfer function of the load system as

$$G(S) = \frac{\omega_n{}^2}{S^2 + 2\omega_n\zeta S + \omega_n{}^2} \tag{1}$$

where $\omega_n$ is the undamped natural frequency, $\zeta$ is the damping ratio, and $S$ is the Laplace operator.

The unit impulse response $y(t)$ of an underdamped second-order system is

$$y(t) = \frac{\omega_n}{\sqrt{1-\zeta^2}} e^{-\zeta\omega_n t} \sin(\omega_d t) \ (t \geq 0) \tag{2}$$

where $\omega_d = \omega_n \sqrt{1-\zeta^2}$, $\omega_d$ is the damping natural frequency.

The input shaper is a set of pulse trains. For a Delta robot, if there is an input shaper with $n$ pulses, it can be calculated by convolution, which can be expressed in the time domain as

$$f(\mathrm{t}) = \sum_{i=1}^{n} A_i \delta(t - t_i) , \ 0 \leq t_i < t_{i+1}, \ A_i > 0 \tag{3}$$

where $\delta(t)$ is the Dirac function (unit impulse function), $A_i$ is the amplitude of the $i$-th pulse, and $t_i$ is the time of the $i$-th pulse. After Laplace transformation, the above formula can be expressed in the frequency domain as

$$F(S) = \sum_{i=1}^{n} e^{-t_i s} \tag{4}$$

To ensure that the amplitude of the gain before and after shaping is the same, determine the amplitude as

$$\sum_{i=1}^{n} A_i = 1 \tag{5}$$

When the amplitudes are all positive, that is, $A_i > 0$, it is said that there is no overshoot at this time. Shaping controllers are designed to implant time delays into the controlled system. To improve the response speed, it is necessary to shorten the response time as much as possible so that the first pulse is realized at zero time; at this time, $t_1 = 0$. If input shaping control is to be performed, $t_i > 0$ should be set [19].

The input pulse sequence $f(t)$ shaped in the time domain is convolved with the original unit impulse response $y(t)$, and the model of the obtained response $Y(t)$ after time $t$ $(t > t_n)$ is

$$\begin{aligned}
Y(t) &= \sum_{i=1}^{n} A_i y(t - t_i) \\
&= \sum_{i=1}^{n} A_i \left[ \frac{\omega_n}{\sqrt{1 - \zeta^2}} e^{-\zeta\omega_n(t - t_i)} \sin(\omega_d t - \omega_d t_i) \right] \\
&= \frac{\omega_n}{\sqrt{1 - \zeta^2}} e^{-\zeta\omega_n t} [A(\omega_n, \zeta) \sin(\omega_d t) - B(\omega_n, \zeta) \cos(\omega_d t)] \\
&= \frac{\omega_n}{\sqrt{1 - \zeta^2}} e^{-\zeta\omega_n t} C(\omega_n, \zeta) \sin(\omega_d t + \gamma)
\end{aligned} \tag{6}$$

where

$$\begin{aligned}
A(\omega_n, \zeta) &= \sum_{i=1}^{n} A_i e^{\zeta\omega_n t_i} \cos(\omega_d t_i) \\
B(\omega_n, \zeta) &= \sum_{i=1}^{n} A_i e^{\zeta\omega_n t_i} \sin(\omega_d t_i) \\
C(\omega_n, \zeta) &= \sqrt{A(\omega_n, \zeta)^2 + B(\omega_n, \zeta)^2} \\
\tan\gamma &= -\frac{B(\omega_n, \zeta)}{A(\omega_n, \zeta)}
\end{aligned} \tag{7}$$

The amplitude ratio of the shaped impulse response $Y(t)$ and the original impulse response $y(t)$ after $t_n$ time is used as the performance index of the input shaping, and the ratio is called the residual vibration percentage, which is defined as

$$V(\omega_n, \zeta) = e^{-\zeta\omega_n t_n} C(\omega_n, \zeta) \tag{8}$$

where $e^{-\zeta\omega_n t_n}$ represents the introduction of a time delay $t_n$ in the shaped response. This ratio reflects the suppression effect of the residual vibration, and the design goal of the input shaper is to make $V \approx 0$ [13].

### 4.2. Construction of a Single-Modal Input Shaper

The types of single-modal input shapers mainly include the ZV (Zero-Vibration) input shaper, ZVD (Zero-Vibration and Derivative) input shaper, and EI (Extra-Insensitive) input shaper.

4.2.1. ZV Input Shaper

The ZV input shaper is also known as a dual-pulse input shaper because it becomes zero residual system vibration when the system's natural frequency and damping ratio are accurately calculated. Take $n = 2$ as the double pulse (ZV) input shaper. The ZV input shaper shapes the pulse signal into two pulse signals with amplitudes $A_1$ and $A_2$. After the first pulse signal $A_1$ generates a dynamic response in the system, the second pulse signal $A_2$ is sent out at interval $\Delta T$. If the two pulse signals are equal in magnitude and opposite in direction, the signals can be canceled out to achieve vibration suppression. The shaping process of the ZV shaper under the impulse response is shown in Figure 9.

The ZV input shaper can be described in the time domain as

$$F(t) = A_1\delta(t) + A_2\delta(t - t_2) \tag{9}$$

It can be described in the frequency domain as

$$F(S) = \sum_{i=1}^{2} A_i e^{-t_i s} \tag{10}$$

Let Formula (8) be equal to 0, and set the equation system as

$$\begin{cases} A(\omega_n, \zeta) = 0 \\ B(\omega_n, \zeta) = 0 \\ \sum\limits_{i=1}^{2} A_i = 1 \end{cases} \tag{11}$$

The ZV input shaper can be obtained:

$$\mathrm{ZV} = \begin{bmatrix} A_i \\ t_i \end{bmatrix} = \begin{bmatrix} \frac{1}{1+K} & \frac{K}{1+K} \\ 0 & \frac{T}{2} \end{bmatrix}, \tag{12}$$

When $T$ is the vibration period, $T = 2\pi/\omega_d$, $K = e^{-\pi\zeta\omega_n/\omega_d}$.

4.2.2. ZVD Input Shaper

In practice, the ZV input shaper has high requirements for the accuracy of mathematical modeling. In theory, when accurate $\omega_n$ and $\zeta$ are obtained, the ZV input shaper can completely eliminate the residual vibration. If there are certain uncertain factors in the working process of the robot that lead to changes in system parameters, the pulse input to the shaper cannot be completely offset from the pulse of the original signal, resulting in a poor vibration suppression effect. Then, high-order input shapers can be used, which are robust to modeling errors [25].

ZVD is an input shaper with three pulse signals, which can be described in the frequency domain as

$$F(S) = \sum_{i=1}^{3} A_i e^{-t_i s} \tag{13}$$

Calculate the partial derivative of $A(\omega_n, \zeta)$ and $B(\omega_n, \zeta)$ at $\omega = \omega_n$ and make it equal to 0, and set the equation system as

$$\begin{cases} \frac{\partial A(\omega,\zeta)}{\partial \omega}|\omega_n = 0 \\ \frac{\partial B(\omega,\zeta)}{\partial \omega}|\omega_n = 0 \\ A(\omega_n, \zeta) = 0 \\ B(\omega_n, \zeta) = 0 \\ \sum\limits_{i=1}^{3} A_i = 1 \end{cases} \tag{14}$$

The ZVD shaper can be obtained:

$$\text{ZVD} = \begin{bmatrix} A_i \\ t_i \end{bmatrix} = \begin{bmatrix} \frac{1}{D_1} & \frac{2K}{D_1} & \frac{K^2}{D_1} \\ 0 & \frac{T}{2} & T \end{bmatrix} \tag{15}$$

$$D_1 = 1 + 2K + K^2 \tag{16}$$

ZVD is more robust than the ZV input shaper, but the time delay is half a cycle longer than the ZV input shaper. As with all input shapers, a trade-off must be made between the increased robustness and the signal time lag caused by the increased shaper duration.

### 4.2.3. ZVDD Input Shaper

To further increase the robustness, this process can be repeated with higher-order derivatives with respect to the frequency, whose second derivative (ZVDD shaper) can be expressed as

$$\text{ZVDD} = \begin{bmatrix} A_i \\ t_i \end{bmatrix} = \begin{bmatrix} \frac{1}{D_2} & \frac{3K}{D_2} & \frac{3K^2}{D_2} & \frac{K^3}{D_2} \\ 0 & \frac{T}{2} & T & \frac{T}{3} \end{bmatrix} \tag{17}$$

$$D_2 = 1 + 3K + 3K^2 + K^3 \tag{18}$$

### 4.2.4. EI Input Shaper

The input shapers discussed above are all designed under the constraints of the residual vibration percentage $V = 0$. However, in actual situations, the natural frequency and damping ratio of the system cannot be accurately estimated, and it is difficult to obtain an accurate model, which will lead to poor vibration suppression effects, and the use of high-order input shapers will double the signal time delay.

If the residual vibration percentage $V = 0$ is not required but at the system frequency $\omega = \omega_n$, $V = V_\text{exp}(V_\text{exp} > 0)$, $\frac{\partial V(\omega,\zeta)}{\partial \omega}|\omega_n = 0$. When the system frequency is $\omega \neq \omega_n$, $V = 0$. Therefore, the input shaper can guarantee the residual vibration percentage $V \leq V_\text{exp}$ of the system. Such an input shaper that allows the percentage of residual vibration to remain below a certain value is called an EI shaper [26]. The EI shaper has three pulse signals, which can be described in the frequency domain as

$$F(S) = \sum_{i=1}^{3} A_i e^{-t_i s} \tag{19}$$

The EI input shaper can be expressed as

$$\text{EI} = \begin{bmatrix} A_i \\ t_i \end{bmatrix} = \begin{bmatrix} A_1 & A_2 & A_3 \\ 0 & t_2 & T \end{bmatrix} \tag{20}$$

When the damping ratio is $\zeta \neq 0$, the pulse signal $A_i$ and pulse time $t_i$ of the EI input shaper can be expressed as [19]

$$\begin{aligned} A_1 &= 0.2479 + 0.2496V_\text{exp} + 0.8001\zeta + 1.233V_\text{exp}\zeta + 0.496\zeta^2 + 3.173V_\text{exp}\zeta^2 \\ A_3 &= 0.2515 + 0.2147V_\text{exp} - 0.8325\zeta + 1.415V_\text{exp}\zeta + 0.8518\zeta^2 + 4.9V_\text{exp}\zeta^2 \\ A_2 &= 1 - (A_1 + A_3) \end{aligned} \tag{21}$$

$$t_2 = T\left(0.4999 + 0.46159V_\text{exp}\zeta + 4.26169V_\text{exp}\zeta^2 + 1.75601V_\text{exp}\zeta^3 + 8.57843V_\text{exp}^2\zeta - 108.644V_\text{exp}^2\zeta^2 + 336.898V_\text{exp}^2\zeta^3\right) \tag{22}$$

The EI input shaper has the same signal time delay as the ZVD input shaper, but it allows a certain percentage of residual vibration, so it is more robust.

### 4.3. Simulink Implementation of a Single-Modal Input Shaper

According to the parameters of the required pulse amplitude $A_i$ and pulse time $t_i$ calculated above, the input shapers of ZV, ZVD, ZVDD, and EI are constructed in Simulink in Figure 11.

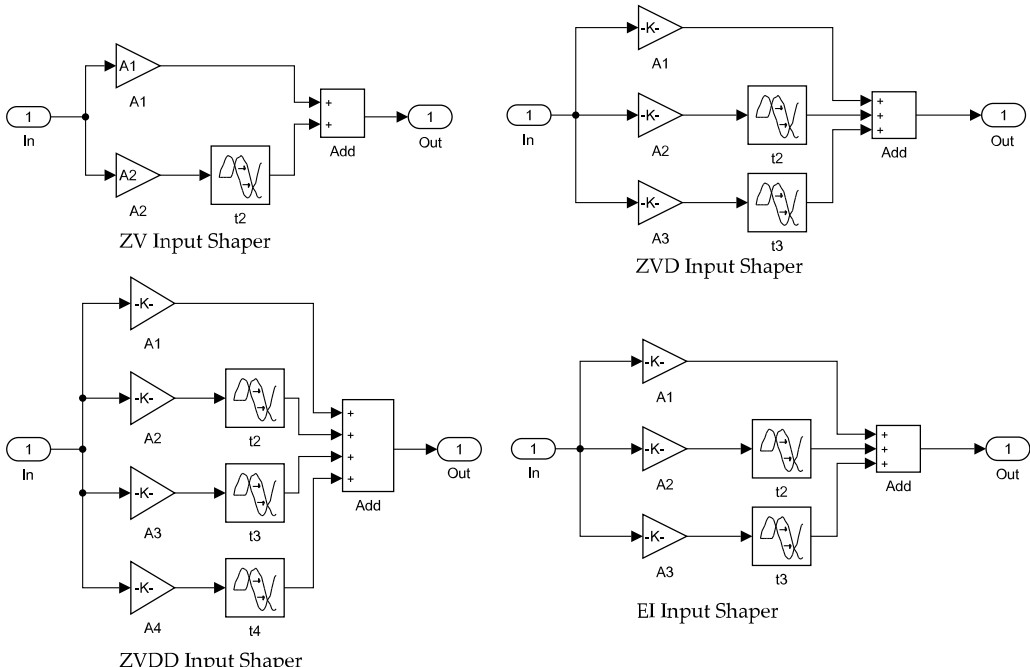

**Figure 11.** Input shaper module.

### 4.4. Multimodal Input Shaper

When the high-order modal has a great influence on the residual vibration of the system, the single-modal input shaper cannot effectively suppress the residual vibration of the multi-modal system, and a multi-modal input shaper is required [27]. Multiple single-modal input shapers are convolved to form a multi-modal input shaper, which has the advantage of suppressing the residual vibration generated by multiple modals of the system at the same time and has good vibration suppression for systems with multiple modal effects and robustness. The principle of multimodal input shaping is shown in Figure 12.

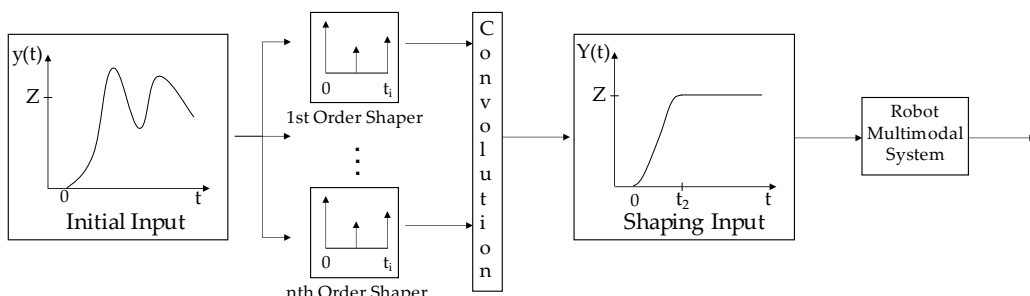

**Figure 12.** Multimodal input shaping control principle.

### 4.5. Design of a Dual-Modal Input Shaper

According to the modal analysis in the previous section, the first two order modal vibration shapes of each position point of the Delta robot have a great influence on the residual vibration of its end effector, and the frequency bandwidth and amplitude of the two modes are quite different, so the vibration suppression ability of the single-modal

input shaper to the robot is reduced. Therefore, a dual-modal input shaper is designed to overcome these drawbacks. The dual-modal input shaper is built by convolving two single-modal input shapers. For example, the first-order modal of the system is 10 Hz, and the second-order modal is 15 Hz; the input shaper of each order modal can be described as

$$\text{ZV} = \begin{bmatrix} A_i \\ t_i \end{bmatrix} = \begin{bmatrix} A_1 & A_2 \\ 0 & t_2 \end{bmatrix} (10 \text{ Hz } shaper) \tag{23}$$

$$\text{ZVD} = \begin{bmatrix} A_i \\ t_i \end{bmatrix} = \begin{bmatrix} A_1 & A_2 & A_3 \\ 0 & t_2 & t_3 \end{bmatrix} (15 \text{ Hz } shaper) \tag{24}$$

where the ZV input shaper is used for the first-order modal shaping and the ZVD input shaper is used for the second-order modal shaping so that the ZV-ZVD dual-mode input shaper can be established.

Convolve the single-mode input shaper given by Formulas (23) and (24), which generate a ZV-ZVD shaper, as shown in Formula (25):

$$\text{ZV} - \text{ZVD} = \begin{bmatrix} A_i \\ t_i \end{bmatrix} = \begin{bmatrix} A_1 & A_2 & A_3 & A_4 \\ t_1 & t_2 & t_3 & t_4 \end{bmatrix} \tag{25}$$

According to the above, this paper establishes six dual-mode input shapers for the Delta robot system, namely ZV-ZV, ZVD-ZVD, EI-EI, ZV-ZVD, ZV-EI, and ZVD-EI. Since the ZVDD input shaper response time lag is too high, ZVDD is not used for the design of the dual-modal input shaper, and it is only used as a reference.

## 5. Simulation and Analysis

### 5.1. Design of a Simulink Block Diagram

According to the modal analysis in the third section, the average value of the natural frequency of the first two modes at each position point is selected to establish the input shaper; $\omega_{n1} = 39.36$ and $\omega_{n2} = 40.45$ can be obtained, and the damping ratio is $\zeta = 0.05$. Substitute the above parameters into the second-order transfer function of the system (1). Taking the unit step signal as the input signal of the system, the six dual-mode input shapers established in the previous section are simulated and analyzed, and the simulation time is 0.5 s. The block diagram of the dual-mode input shaper is shown in Figure 13. The simulation results are shown in Figure 14.

It can be seen from Figure 14 that the ZV-ZV dual-modal input shaper has the best vibration suppression performance, but the ZV-ZV dual-modal input shaper requires precise system modal parameters. However, the modal parameters of the Delta robot will change continuously during the movement in the workspace and there will be errors between the simulation and the actual system, so the robustness of the dual-modal input shaper is particularly important.

### 5.2. Robustness Analysis of the Dual-Modal Input Shaper

Robustness is a key index to evaluate the ability of the input shaper to suppress residual vibration, so the designed dual-modal input shaper can be used in practical engineering only if it has good robustness. In order to analyze the robustness of each input shaper, a certain error value is given to the modal parameters in this paper, and then the vibration suppression performance of each input shaper in the actual work of the Delta robot is simulated.

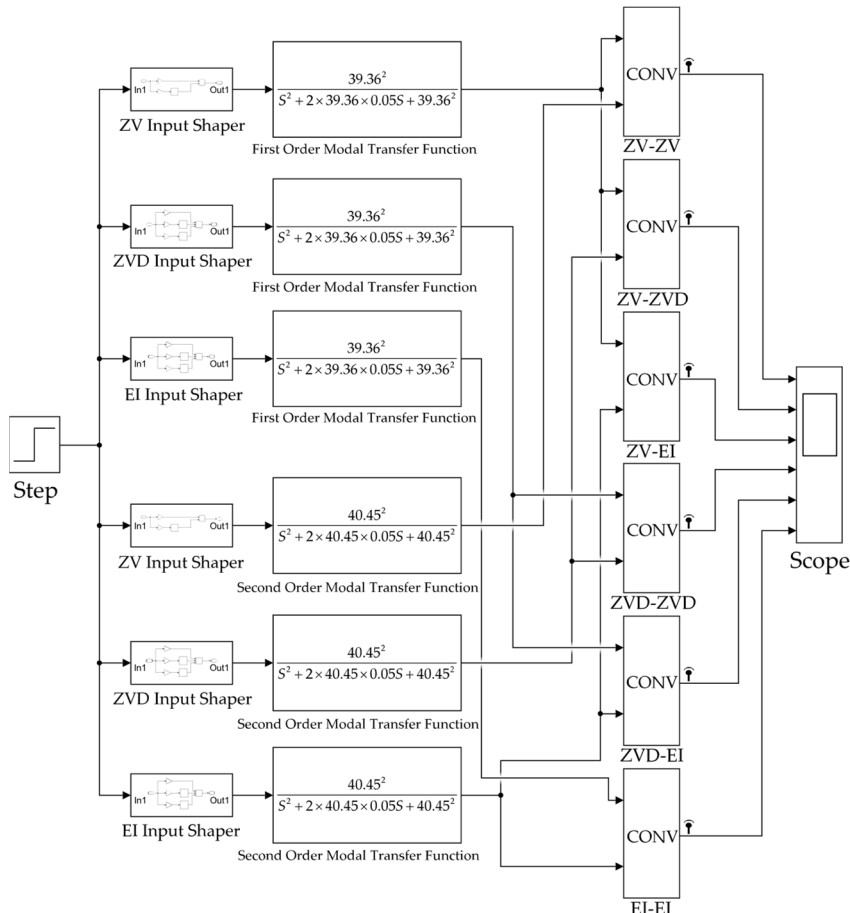

**Figure 13.** Block diagram construction of dual-modal input shapers.

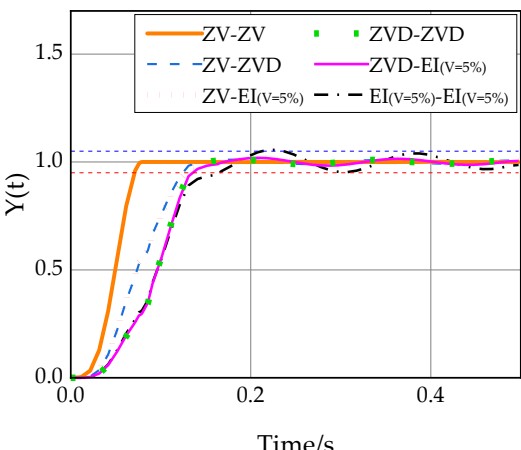

**Figure 14.** Unit step response of each dual-mode input shaper.

It can be seen from Figure 15 that all types of dual-modal input shapers can effectively suppress the residual vibration of the first two order modals. Among them, the ZV-ZV input shaper has the fastest response speed and no overshoot, but when the system modal parameter error increases, its residual-vibration-suppression effect gradually becomes worse, and its robustness is poor, as shown in Figure 15a–d. The EI-EI input shaper can also have good robustness when the system modal parameter error increases, but at the cost of increasing the adjustment time. Taken together, the ZVD-EI input shaper can significantly

reduce the residual vibration of the first two order modals of the Delta robot and it has good robustness and a short adjustment time.

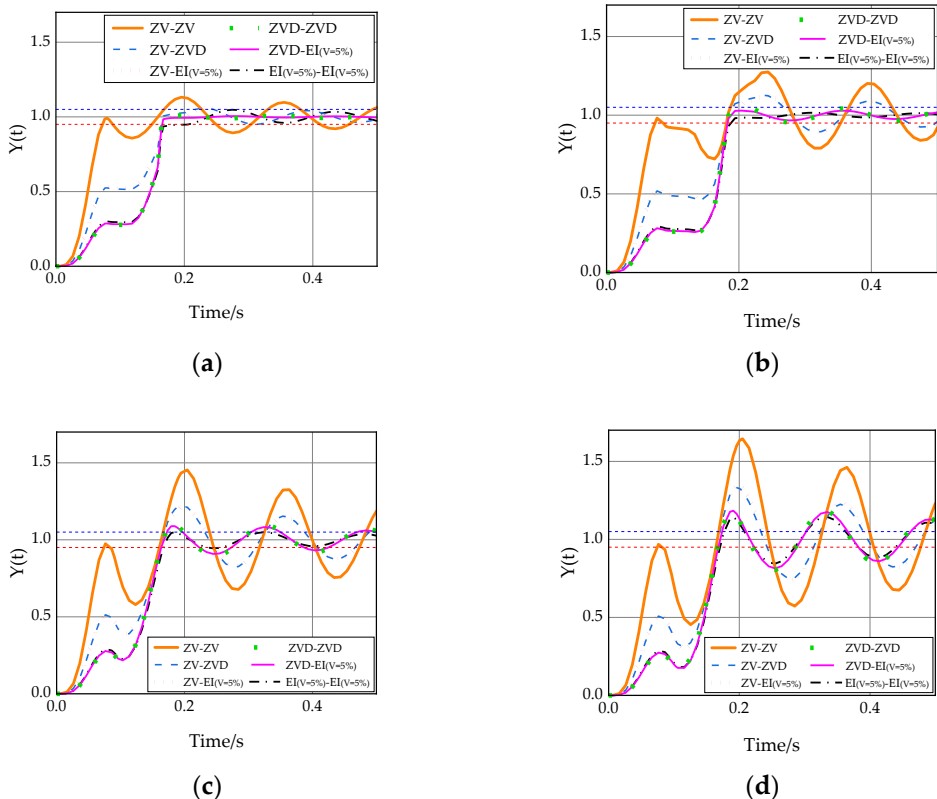

**Figure 15.** Unit step response of each dual-modal input shaper when the system modal parameters have error values: (**a**) response when the system modal parameter error value is 5%; (**b**) response when the system modal parameter error value is 10%; (**c**) response when the system modal parameter error value is 15%; (**d**) response when the system modal parameter error value is 20%.

### 5.3. Verification of the ZVD-EI Dual-Modal Input Shaper

The natural frequency parameter of the single-modal input shaper in this section is the average natural frequency of the first-order modal as $\omega_{n1} = 39.36$. According to Figure 5, it can be seen that the natural frequency difference of the first two order modals of the Delta robot at each position point does not exceed 10% of the average value. Therefore, the modal parameter error value is set to 10% for simulation analysis, that is, $\omega_{n1} = 35.424$, $\omega_{n2} = 36.405$. Taking the unit step signal as the input signal of the system and the simulation time is 0.5 s, the simulation results are shown in Table 5. It can be seen from Table 5 that among single-modal input shapers, the ZV input shaper has the largest overshoot, the longest adjustment time, and the worst robustness; the ZVDD input shaper has the smallest overshoot but the longest adjustment time. Among the dual-modal input shapers, the ZVD-ZVD input shaper has the shortest adjustment time but slightly higher overshoot; the ZVD-EI input shaper has the smallest overshoot, and the adjustment time is similar to the ZVD-ZVD input shaper. On the whole, the ZVD-EI input shaper has the advantages of small overshoot and a short adjustment time, and its robustness and vibration suppression ability are the best.

**Table 5.** Performance of each input shaper at 10% error value.

| Input Shaper | Overshoot/% | Adjustment Time/s |
|---|---|---|
| ZV | 12.66 | 0.606 |
| ZVD | 2.28 | 0.156 |
| ZVDD | 0.04 | 0.244 |
| EI$_{(V=5\%)}$ | 2.05 | 0.166 |
| ZV-ZV | 11.33 | 0.606 |
| ZV-ZVD | 11.53 | 0.682 |
| ZV-EI$_{(V=5\%)}$ | 12.22 | 0.687 |
| ZVD-ZVD | 1.35 | 0.156 |
| ZVD-EI$_{(V=5\%)}$ | 0.16 | 0.158 |
| EI$_{(V=5\%)}$-EI$_{(V=5\%)}$ | 3.92 | 0.170 |

In order to further verify the superiority of the ZVD-EI dual-modal input shaper, it is compared with the ZVD single-modal input shaper commonly used in industrial robot vibration suppression [19–21], and the robustness and vibration suppression effect of the traditional ZVD single-modal input shaper and the ZVD-EI dual-modal input shaper proposed in this paper are mainly compared. As shown in Figure 16a, when there is no modal parameter error value in the system, compared with the traditional ZVD single-modal input shaper, the adjustment time of the ZVD-EI dual-modal input shaper is reduced by 24.14% and the amplitude is smaller, which has better vibration suppression performance. As shown in Figure 16c, when the modal parameter error value is 10%, the adjustment time of ZVD-EI dual-modal input shaper is 17.2% shorter than that of ZVD input shaper. With a certain modal parameter error value, compared with the traditional ZVD input shaper, the ZVD-EI input shaper still has good robustness and residual vibration suppression performance, as shown in Figure 16b–e.

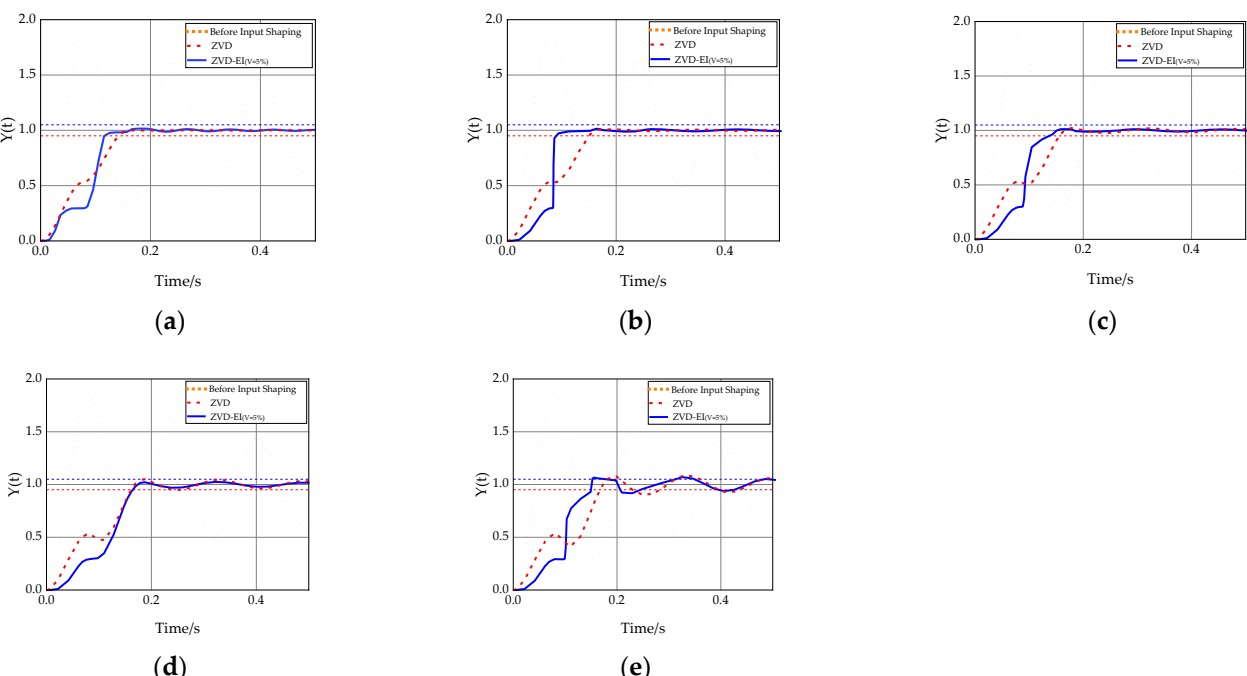

**Figure 16.** Unit step response of the ZVD-EI dual-modal input shaper and the ZVD single-modal input shaper: (**a**) response when the system modal parameter error value is 0; (**b**) response when the system modal parameter error value is 5%; (**c**) response when the system modal parameter error value is 10%; (**d**) response when the system modal parameter error value is 15%; (**e**) response when the system modal parameter error value is 20%.

*5.4. Selection Principles of Dual-Modal Input Shapers*

For the six dual-modal input shapers established above, the ZVD-EI dual-modal input shaper is most suitable for application in the Delta robot system established in this paper and has good robustness and residual vibration suppression performance.

The dual-modal input shaper designed in this paper can also be applied to other robots with multiple dominant modal and industrial equipment controlled by controllers and motors. When the vibration frequency band of the first-order modal of the system is narrow and the vibration frequency band of the second-order modal is wide, the ZV-EI or ZV-ZVD dual-modal input shaper can be used for residual vibration suppression. When the vibration frequency band of the first-order modal of the system is wide and the vibration frequency band of the second-order modal is narrow, the EI-ZV or ZVD-ZV dual-modal input shaper can be used for residual vibration suppression. Among the above six input shapers, the ZVD-EI dual-modal input shaper has the best robustness and comprehensive performance. The ZV-ZV dual-modal input shaper has the shortest adjustment time but the worst robustness and is suitable for systems where the vibration frequency bands of each modal of the system are narrow. Therefore, the most suitable dual-modal input shaper can be selected for residual vibration suppression according to the vibration frequency bandwidth of each order modal of the system, the expected vibration amplitude, and the adjustment time.

## 6. Conclusions

(1)  A design method of a dual-modal input shaper was presented to solve the poor effect of the traditional input shaper on residual vibration suppression caused by the change of the dynamic characteristics of the Delta robot in its workspace. Through simulation analysis, it is found that among the six dual-modal input shapers established in this paper, the ZVD-EI dual-modal input shaper has the best robustness and residual vibration suppression performance.

(2)  The simulation verification in Simulink shows that compared with the traditional ZVD single-modal input shaper, when the system modal parameters are accurate, the adjustment time of the ZVD-EI dual-mode input shaper is reduced by about 24%; when the parameter error value is 10%, the adjustment time of the ZVD-EI dual-modal input shaper is reduced by about 17%. Compared with other input shapers, the ZVD-EI dual-modal input shaper has better robustness and comprehensive performance and can effectively reduce the residual vibration of the Delta robot.

(3)  By analyzing the usage principles of the established dual-modal input shaper, different dual-modal input shapers in different robots and industrial equipment can be selected for residual vibration suppression according to their system characteristics. In the future, the ZVD-EI dual-modal input shaper algorithm will be applied to the Delta robot prototype to further improve its residual vibration suppression performance and make this method apply better to actual engineering.

**Author Contributions:** Z.G.: conception of the study, proposition of the theory and method, supervision; J.Z.: literature search, figures, data collection, manuscript preparation and writing; P.Z.: programming, testing of existing code components. All authors have read and agreed to the published version of the manuscript.

**Funding:** This research was funded by the Liaoning Provincial Education Department Project (Grant No. LJKZ0114).

**Data Availability Statement:** Not applicable.

**Conflicts of Interest:** The authors declare no conflict of interest.

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
