# Peer review of "Research on the Residual Vibration Suppression of Delta Robots Based on the Dual-Modal Input Shaping Method"

_actuators, doi:10.3390/act12020084_

Round 1

Reviewer 1 Report

The paper presents a vibration analysis of a Delta robot and an approach for the vibration suppression on that robot. The topics of the paper are interesting and relevant for the journal. However, the quality of the paper is overall low, and the work is not suitable for publication in the present form. The following aspect should be carefully considered in order to improve the overall quality and readability of the manuscript:

1) The main contributions of the paper are not clear. The novelty and originality of the work should be highlighted, especially with respect to similar works on the topic.

2) The overall quality and style of the paper should be improved. All figures have low quality and cannot be read easily. All the figures should be replaced with high quality images. I suggest using vector images.

3) The geometrical and dynamical characteristics of the robot under study are not reported in the manuscript. It is not clear how the modal analysis has been conducted. It would be impossible to replicate the numerical simulations.

4) It is not clear to me what the authors did in Section 4. The mathematical formulation lacks rigor. That part of the paper should be completely rewritten and clarified. How the parallel robot was simulated in Simulink? Did the authors implement a flexible dynamical model of that robot?

5) The results of the paper are not clear. The authors should highlight what are the main results of the paper, and how the proposed approach performs with respect to alternative approaches available in the literature.

6) The literature review should be improved. Many recent papers can be found in the literature dealing with parallel robots. The following suggested references should be considered to improve the state-of-the-art section:

A trajectory planning approach for Delta robots considering both motion smoothness and dynamic stress. Journal of Mechanisms and Robotics, 1-54, 2022.

Enhancing energy efficiency of a 4-DOF parallel robot through task-related analysis. Machines8(1), 10, 2020.

Research on intelligent vibration suppression control of high-speed lightweight Delta robot. Journal of Vibration and Control28(21-22), 3042-3057, 2022.

An energy-efficient approach for 3D printing with a Linear Delta Robot equipped with optimal springs. Robotics and Computer-Integrated Manufacturing67, 102045, 2021.

Reviewer 2 Report

This paper presents a method to attenuate a Delta robot's residual vibration, considering the end-effector's dominant mode. The proposed approach is validated by means of simulation results. 

  1. Abstract: it needs to be evident to identify the novel contribution. What is the original contribution regarding previous research studies reported in the literature?

  2. Introduction:  

    1. The literature review can be improved: two approaches were reviewed (passive vibration control and active vibration control) applied to the Delta Robot. Nevertheless, alternative methods have been used to parallel robots to suppress vibration. The literature review should be improved to highlight the novel contribution of the present research study. Some references are suggested for the Delta robot:

Zhao, R., Wu, L., & Chen, Y. H. (2020). Robust control for nonlinear delta parallel robot with uncertainty: an online estimation approach. IEEE Access, 8, 97604-97617.

Zu, Q., Liu, Q., & Wu, J. (2021). Dynamic Pick and Place Trajectory of Delta Parallel Manipulator. In International Conference on Human Centered Computing (pp. 1-11). Springer, Cham.

Dai, Z., Sheng, X., Hu, J., Wang, H., & Zhang, D. (2015, August). Design and implementation of Bézier curve trajectory planning in DELTA parallel robots. In International Conference on Intelligent Robotics and Applications (pp. 420-430). Springer, Cham.

Mirz, C., Hüsing, M., Takeda, Y., & Corves, B. (2022). Low Cost Delta Robot for the Experimental Validation of Frame Vibration Reduction Methods. In Symposium on Robot Design, Dynamics and Control (pp. 332-339). Springer, Cham.

Wu, L., Zhao, R., Li, Y., & Chen, Y. H. (2020). Optimal design of adaptive robust control for the delta robot with uncertainty: fuzzy set-based approach. Applied Sciences, 10(10), 3472.

Moreover, it is recommended to carry out a literature review focused on parallel robots and vibration control.

  1. In my opinion, the main contribution is “input shaping control” of section 4. Nevertheless, this approach was not introduced at the beginning of the manuscript.

  1. The figure's quality is so poor: improve the quality. 

  2. Elastodynamic model: It is not evident to understand this model: The results obtained from Ansys were presented in this section. Nevertheless, the inertia and stiffness parameters were not defined. Moreover, the dimensions of the manipulator were not defined. Was this model based on a previous contribution?

  3. Results: 

    1. Could the “input Shaping Control” be applied on another robot? or it only works on Delta-robot? Why?

    2. It is not evident to evaluate how the proposed control approach enhances the elastodynamic performance of the robot. It is necessary to compare to at least another previous approach proposed in the literature.

    3. The simulation was carried out with a simplified model of the manipulator (according to Fig. 11). The dynamic simulation was computed with the complete model of the manipulator. It harms the quality of the results. 

  4. Conclusions are not supported by the results.

Round 2

Reviewer 1 Report

The paper has been somehow improved with respect to the previous version. However, the mathematical modelling is still not clear to me. More in detail:

The overall quality and style of the paper should be improved. All figures have low quality and cannot be read easily. All the figures should be replaced with high quality images. I suggest using vector images.

Not sufficient details are given about the model of the robot in Ansys.

It is not clear to me what the authors did in Section 4. The mathematical formulation lacks rigor. That part of the paper should be completely rewritten and clarified. How the parallel robot was simulated in Simulink? Did the authors implement a flexible dynamical model of that robot?

The results of the paper are not clear. The authors should highlight what are the main results of the paper, and how the proposed approach performs with respect to alternative approaches available in the literature.

Reviewer 2 Report

The authors did not answer all inquiries pointed out in the first review:

"2. The literature review can be improved: two approaches were reviewed (passive vibration control and active vibration control) applied to the Delta Robot. Nevertheless, alternative methods have been used to parallel robots to suppress vibration. The literature review should be improved to highlight the novel contribution of the present research study. Some references are suggested for the Delta robot"

"Could the “input Shaping Control” be applied on another robot? or it only works on Delta-robot? Why?"

"It is not evident to evaluate how the proposed control approach enhances the elastodynamic performance of the robot. It is necessary to compare to at least another previous approach proposed in the literature. "

"The simulation was carried out with a simplified model of the manipulator (according to Fig. 11). The dynamic simulation was computed with the complete model of the manipulator. It harms the quality of the results. "

"Conclusions are not supported by the results. "

Round 3

Reviewer 1 Report

The paper has been somehow improved with respect to the previous version. Some additional comments:

Is it not clear how the frequencies of Figure 1 have been computed. Is it possible to provide an acceleration signal (of the more relevant points of the robot) in time and frequency domain? I suggesting adding more graphs to improve the quality of the results.

Please provide graphs of the robot position, velocity, and acceleration both in the joint space and in the Cartesian space.

It would be interesting to perform an analysis on a different additional trajectory. 

Figures 3, 4, 5, 6, and 12 should still be improved.

Reviewer 2 Report

The authors included all the corrections and comments in this new version of the manuscript. 

Author Response

Dear reviewer:

The co-authors and I would like to thank you for the time and effort spent in reviewing the manuscript. Thanks again for your guidance and comments.